# Real-Time Hand Gesture Recognition in Clinical Settings: A Low-Power FMCW Radar Integrated Sensor System with Multiple Feature Fusion

**DOI:** 10.3390/s25134169

**Published:** 2025-07-04

**Authors:** Haili Wang, Muye Zhang, Linghao Zhang, Xiaoxiao Zhu, Qixin Cao

**Affiliations:** 1The State Key Laboratory of Mechanical System and Vibration, Shanghai Jiao Tong University, Shanghai 200240, China; sjtu_wanghaili@sjtu.edu.cn; 2SJTU Paris Elite Institute of Technology, Shanghai Jiao Tong University, Shanghai 200240, China; zhangmuye@sjtu.edu.cn; 3School of Computer Science, Shanghai Jiao Tong University, Shanghai 200240, China; zhanglinghao@sjtu.edu.cn; 4Student Innovation Center, Shanghai Jiao Tong University, Shanghai 200240, China; ttl@sjtu.edu.cn

**Keywords:** hand gesture recognition, millimeter-wave radar, deep learning, clinical edge computing, multiple feature fusion, real-time, medical sensor systems, IoT

## Abstract

Robust and efficient contactless human–machine interaction is critical for integrated sensor systems in clinical settings, demanding low-power solutions adaptable to edge computing platforms. This paper presents a real-time hand gesture recognition system using a low-power Frequency-Modulated Continuous Wave (FMCW) radar sensor, featuring a novel Multiple Feature Fusion (MFF) framework optimized for deployment on edge devices. The proposed system integrates velocity profiles, angular variations, and spatial-temporal features through a dual-stage processing architecture: an adaptive energy thresholding detector segments gestures, followed by an attention-enhanced neural classifier. Innovations include dynamic clutter suppression and multi-path cancellation optimized for complex clinical environments. Experimental validation demonstrates high performance, achieving 98% detection recall and 93.87% classification accuracy under LOSO cross-validation. On embedded hardware, the system processes at 28 FPS, showing higher robustness against environmental noise and lower computational overhead compared with existing methods. This low-power, edge-based solution is highly suitable for applications like sterile medical control and patient monitoring, advancing contactless interaction in healthcare by addressing efficiency and robustness challenges in radar sensing for edge computing.

## 1. Introduction

The integration of advanced sensor systems in medical applications has become increasingly vital in modern healthcare, enabling continuous monitoring and real-time data collection to enhance patient outcomes. These systems, which often comprise multiple interconnected sensors, facilitate the gathering and processing of critical health-related data, thereby providing healthcare professionals with valuable insights for diagnostics and personalized treatments. Among the various modalities of integrated sensor systems, contactless hand gesture recognition has emerged as a promising solution for enhancing human-computer interaction (HCI) in medical environments, particularly in the context of remote patient monitoring and telemedicine [1].

Traditional HCI methods, such as vision-based cameras or wearable sensors, face significant challenges, including lighting dependency, occlusion issues, and the need for intrusive hardware [2]. In contrast, millimeter-wave (mmWave) radar technology offers unique advantages, such as illumination independence and high-precision motion tracking, making it an ideal candidate for medical applications where hygiene and non-invasiveness are paramount [3,4,5]. Recent advancements in FMCW mmWave radar have further enhanced the capabilities of gesture recognition systems, allowing for sophisticated time-frequency feature extraction that is crucial for accurate patient monitoring [6,7].

Despite these advancements, most systems rely on Convolutional Neural Network (CNN)-based micro-Doppler analysis, often overlooking critical spatial features like angle-of-arrival (AoA) and range-depth fusion [8]. High-resolution angle estimation techniques, while promising, often suffer from computational complexity that limits their applicability in real-time medical scenarios [9,10]. Our research addresses these limitations by introducing an innovative L-shaped antenna array configuration combined with Digital Beamforming (DBF) technology, which enables efficient construction of multi-feature maps that integrate range, velocity, and AoA information with minimal computational overhead [10].

The challenge of temporal modeling in gesture recognition is particularly relevant in medical contexts, where accurate gesture segmentation can significantly impact patient interaction with healthcare systems. To address this, we propose a novel two-stage detection mechanism that features initial coarse frame localization followed by optimized trigger determination, enhanced by DBSCAN-based noise reduction and multi-modal radar fusion for improved robustness in diverse medical environments [5,6,11].

This work (Figure 1) makes three key contributions to the field of integrated sensor systems for medical applications:Multi-feature Fusion Framework: We develop a computationally efficient feature fusion method that combines range, velocity, and AoA information through optimized DBF processing, achieving high spatial resolution with minimal computational overhead [10].Hierarchical Recognition Architecture: Our detector-classifier pipeline significantly improves segmentation accuracy while reducing computational latency through intelligent feature processing, making it suitable for real-time medical applications [5,11].Edge-Computing Optimization: We demonstrate practical implementation on resource-constrained edge devices, maintaining real-time performance without compromising recognition accuracy, validated through extensive testing on Raspberry Pi platforms [9,12].

Unlike existing research that prioritizes theoretical accuracy over practical implementation, our framework specifically addresses the critical need for efficient and effective integrated sensor solutions in real-world medical applications, paving the way for enhanced patient monitoring and interaction in healthcare settings.

The organization of this paper is as follows: Section 2 critically reviews recent advancements in radar-based gesture recognition systems and healthcare applications of contactless sensing technology. Section 3 details the proposed system’s methodology, including the FMCW radar hardware configuration, signal processing pipeline, and the novel Multiple Feature Fusion (MFF) framework. Section 4 presents experimental validation through quantitative benchmarks and comparative analysis against state-of-the-art methods. Section 5 discusses practical deployment scenarios and clinical applications. Finally, Section 6 concludes with a summary of contributions to gesture recognition in healthcare settings.

## 2. Related Works

### 2.1. Radar-Based Gesture Recognition Systems

Current radar-based gesture recognition systems (e.g., Table 1) prioritize CNN-driven micro-Doppler analysis but often neglect spatial cues like range and angle [8]. While Multiple-Signal-Classification (MUSIC)/Estimation-of-Signal-Parameters-via-Rotational-Invariance-Techniques (ESPRIT) methods enhance angular resolution, their computational complexity limits real-time deployment [9]. Our work addresses this gap by adopting an L-shaped antenna array with Digital Beamforming (DBF) to fuse range, velocity, and AoA into a low-channel multi-feature map, optimizing Internet of Things (IoT) compatibility without sacrificing spatial fidelity [10]. For temporal modeling, hybrid architectures combining 3D-CNNs (short-term features) and Long Short-Term Memory (LSTMs) (long-term dependencies) outperform single-model approaches [4,8], achieving 96% accuracy in prior studies.

Real-time segmentation poses a significant challenge. Threshold-based detectors often struggle with noise and variability across multiple gestures [11], prompting our two-stage solution: coarse frame localization followed by windowed trigger optimization. To enhance the robustness, we integrate radar and Density-Based Spatial Clustering of Applications with Noise (DBSCAN) based denoising, mitigating interference at the feature level [5]. Unlike works prioritizing accuracy over deployability (e.g., Multiple-Input-Multiple-Output (MIMO) radar with MUSIC [13] or transformer-based systems [14]), our framework emphasizes edge-computing efficiency, validated on Raspberry Pi for low-power IoT scenarios [9]. This balances accuracy, real-time performance, and resource constraints—a gap in existing literature focused on idealized setups.

**Table 1 sensors-25-04169-t001:** Gesture recognition methods comparison.

Reference	Radar Configuration	Feature Fusion	Learning Method	System Latency	Gesture Diversity
[10]	60 GHz, BGT60TR13C	RDM	Shallow CNN	Edge computing, 33.15 ms	12 gestures
[15]	77 GHz, IWR1443	Micro-Doppler Features	Deep CNN	0.9 ms	9 gestures
[16]	60 GHz, -	RDM + RAM	NN + BiLSTM	-	8 gestures
[17]	24.125 GHz, K-MC1	Micro-Doppler Spectrogram	CNN	-	4 gestures
[18]	60 GHz, BGT60TR13C	Time-Domain Maps	CNN	Edge Device Optimization, 1.28 ms	8 gestures

### 2.2. Healthcare Applications

Radar-based gesture recognition has gained significant traction in healthcare applications (Figure 2) due to its non-contact operability and environmental adaptability [19,20,21]. In sterile surgical environments, gesture-controlled systems enable the touchless manipulation of imaging devices or parameter adjustments, effectively mitigating contamination risks during critical procedures [22]. For patient monitoring, radar systems demonstrate dual functionality: the real-time recognition of limb movements (e.g., arm-raising or turning) facilitates fall detection when integrated with vital sign tracking [8], while edge-computed processing ensures timely alerts for nursing interventions. Rehabilitation therapy benefits from spatial-temporal gesture analysis, where motion trajectories are captured to assess motor coordination in patients with mobility impairments, enabling interactive training protocols with real-time feedback via medical cloud platforms [12]. Millimeter-wave radar further addresses communication barriers in clinical settings by translating precise sign language gestures into text or speech through low-power edge devices, enhancing doctor–patient interaction efficiency [23]. Compared with vision-based solutions, radar technology mitigates privacy concerns associated with optical methods [24] while leveraging the penetration capability to monitor occluded scenarios (e.g., movements under blankets) [25]. Edge deployment enhances reliability in high-interference clinical environments (e.g., ICUs), achieving 15.2% enhanced noise resilience over conventional systems. These advancements align with the low-power and sub-50 ms latency characteristics of FMCW radar systems, collectively driving the evolution of intelligent, safety-critical healthcare infrastructures [26,27].

## 3. Materials and Methods

### 3.1. Radar System

#### 3.1.1. System Overview

The frequency-modulated-continuous-wave (FMCW) radar provided by Infineon (Figure 3) operates at a 60 GHz frequency. It has 5 GHz bandwidth max which makes it has a cm-scale range resolution. This radar is equipped with one transmission antenna and three receiving antennas arranged in an L-shaped array. This configuration enables the spatial detection function in both the E-plane and the H-plane (Figure 4). This is crucial for better distinguishing relevant gestures and establishing a more accurate 3D model. As shown in Table 2, which compares the accuracy of different numbers of channels, the setup with three channels achieves the highest accuracy and the best overall performance.

The ESP32 microcontroller acts as an embedded slave unit for real-time signal processing via the SPI interface. Figure 5 depicts the waveform of the FMCW radar in the frequency domain, elaborating on the principles of the transmitter (Tx) and receivers (Rx). The Tx signal is represented by a linearly-frequency-modulated (LFM) waveform, which sweeps through a certain frequency range over time. The Rx signal, on the other hand, is the reflected version of the Tx signal from the target, experiencing a time delay due to the distance of the target.

#### 3.1.2. Signal Processing

In FMCW radar, the transmitted signal is modulated by a set of sine waves with increasing frequency. In Figure 5, by mixing the TX and RX waves, the IF signal is a sine wave with a certain frequency and phase. The frequency of the IF signal is proportional to the range of the object, and the phase change is caused by the relative velocity of the object to the radar. The FMCW waveform parameters govern the range and Doppler velocity resolutions, which can be optimized to enhance hand gesture detection. The resolution of range and Doppler Δr and Δv are listed as follows:(1)$$\begin{array}{c} \begin{array}{c} \Delta r=\frac{c}{2B}=3\;\mathrm{cm} \end{array} \end{array}$$(2)$$\begin{array}{c} \begin{array}{c} \Delta v=\frac{\lambda }{2N(T_{chirp}+\tau_{delay})}=15.5\;\mathrm{cm}/s \end{array} \end{array}$$
where *c* is the speed of light, and *B* is the bandwidth of the FMCW wave shown in Figure 5. λ is the wavelength of the radar. $T_{chirp}$ is the chirp time, which is set to 128 μs, and $\tau_{delay}$ is the delay between the end of the last chirp and the start of the next. The sum of them is set to 505.64 μs and *N* is the number of chirps in one frame is set to 32. The frame rate is set to 24 frames per second (fps) in order to meet the real-time requirement of hand-gesture detection.

The signal processing pipeline for FMCW radar gesture recognition consists of three main stages as illustrated in Figure 6: 2D FFT transformation, phase calibration, and clutter suppression. Each stage is essential for extracting accurate Doppler information from the multichannel radar data.

##### 2D FFT Transformation

As shown in the first stage of Figure 6, the raw signal per frame is first transformed into a matrix form c(p,q,t), where each row contains the samples of a single chirp. Here, *p* and *q* represent the sample index within a chirp and the chirp index within a frame, respectively. The frequency-domain output matrix for frame *t*, obtained through a 2D Fast Fourier Transform (2D FFT), is expressed as follows:(3)$$\begin{array}{c} \begin{array}{c} C_{1D}\left(p,q,t\right)=\sum_{n=1}^{N_{s}}c\left(p,q,t\right)e^{-j2\pi pn/N_{s}} \end{array} \end{array}$$(4)$$\begin{array}{c} \begin{array}{c} C_{2D}\left(p,q,t\right)=\sum_{l=1}^{N_{c}}C_{1D}\left(p,q,t\right)e^{-j2\pi ql/N_{c}} \end{array} \end{array}$$(5)$$\begin{array}{c} \begin{array}{c} RD\left(r,v,t\right)=|C_{2D}\left(p\cdot \Delta r,q\cdot \Delta v,t\right)| \end{array} \end{array}$$

Here, $C_{1D}\left(p,q,t\right)$ and $C_{2D}\left(p,q,t\right)$ represent the complex-valued results of the 1D and 2D Fourier transforms applied to the raw signal, respectively. The range-Doppler map RD(r,v,t) is obtained by computing the magnitude of $C_{2D}\left(p\cdot \Delta r,q\cdot \Delta v,t\right)$, where Δr and Δv denote the range and velocity resolutions, respectively.

##### Calibration

The second stage of Figure 6 illustrates the calibration process. Figure 7 shows the range profiles of antennas Rx2 and Rx3 in the E-plane, with a target 0.5 m from the radar. Figure 6a,b display the amplitude and phase before calibration, showing similar amplitudes but inconsistent phases due to incomplete electromagnetic shielding, which may cause errors in angle estimation without correction. Figure 6c,d display the calibrated range profiles, where amplitude and phase align well at the corner reflector’s location after calibration.

Phase calibration assumes echo signals from a distant target in far-field conditions (target distance ≫ 5 mm wavelength at 60 GHz) are identical across channels when incident from the array’s normal direction, with equal amplitude and phase. A fixed phase offset between receiver antennas can be corrected using zero-degree calibration: place a corner reflector at boresight (zero-degree E-plane and H-plane) at a known distance, calculate the phase difference, and use it as a calibration matrix.

The calibration matrix should be a complex matrix *M* with a shape 2×2 that can be applied to Rx2 and Rx3 complex data pairs $C_{1D}^{23}$ before angle estimation.(6)$$\begin{array}{c} \begin{array}{c} {\left({\hat{C}}_{1D}^{23}\right)}_{2\times 1}={\left(M\right)}_{2\times 2}{\left(C_{1D}^{23}\right)}_{2\times 1} \end{array} \end{array}$$
where ${\hat{C}}_{1D}^{23}$ refers to the calibrated data pair.

The result shows that the amplitude and phase in each plane are well aligned at the location of the corner reflector after the calibration procedure.

##### Clutter Suppression

The final stage of Figure 6 demonstrates the clutter suppression process. The RDM is a crucial representation that combines target range and Doppler frequency information, but it is often contaminated by clutter. Clutter can originate from various sources, such as ground reflections, stationary objects, or other unwanted echoes, which can mask the presence of moving targets and degrade the performance of target detection and identification. In the context of suppressing clutter in the range, and the Doppler map shown in Figure 6, several methods have been developed to address this problem. One commonly used approach is the moving-target-indicator (MTI) technique, which exploits the fundamental difference between target and clutter characteristics. Clutter echoes typically exhibit much larger magnitudes than target echoes but possess zero or very low Doppler frequencies due to their stationary nature. In contrast, moving targets generate relatively high Doppler frequencies. The MTI method leverages this distinction by treating the time-averaged intensity of each pixel in the range-Doppler map as background noise. The clutter matrix can then be subtracted from the RDM with a scaling factor α. The resulting matrix $RD_{MTI}$ represents the RDM with MTI applied:(7)$$\begin{array}{c} \begin{array}{c} RD_{MTI}=RD-\alpha RD_{avg} \end{array} \end{array}$$
where $RD_{avg}$ represents the historical average of the RD matrix. Another method widely adopted in computer vision tasks is the Gaussian Mixture Model-based background segmentation algorithm [28,29]. However, constructing a background model with Gaussian kernels is computationally expensive and challenging to meet real-time requirements.

Therefore, we propose a more efficient clutter extraction method based on adaptive weighting of static objects:(8)$$\begin{array}{c} \begin{array}{c} Mask\left(t\right)=1-\left[\frac{RD\left(r,v,t\right)-RD\left(r,v,t-k\right)}{k\cdot RD\left(r,v,t\right)}\right] \end{array} \end{array}$$(9)$$\begin{array}{c} \begin{array}{c} RD{\left(r,v,t\right)}_{avg}=\frac{1}{k}\sum_{i=t-k}^{t-1}RD\left(r,v,i\right) \end{array} \end{array}$$(10)$$\begin{array}{c} \begin{array}{c} RD{\left(r,v,t\right)}_{new}=RD\left(r,v,t\right)-Mask\left(t\right)\cdot RD{\left(r,v,t\right)}_{avg} \end{array} \end{array}$$

The normalized matrix Mask(t) represents the temporal gradient of each pixel in the RDM between time *t* and time t−k. By element-wise multiplying the Mask(t) with the mean RD matrix calculated from a sequence of frames, static noise can be effectively removed since their corresponding weights in Mask(t) approach zero due to minimal temporal variations. Essentially, this mask serves as an enhanced, adaptive version of the scaling factor α in the conventional MTI algorithm, providing pixel-wise weighting based on temporal changes rather than using a uniform scaling factor. Figure 8 illustrates an example comparing the RDM before and after applying our method.

#### 3.1.3. Multiple Feature Fusion

The principle of azimuth and elevation angle estimation with radar is similar to that of a stereo camera. By calculating the phase angle difference of points with the same position in the complex-valued range-Doppler map, the angle of arrival (AOA) of the object in azimuth and the elevation of object *j* is listed from the phase difference of extracted points in the same positions of complex valued RD spectrums belonging to two receiving antennas as follows:(11)$$\begin{array}{c} \begin{array}{c} \phi_{azimuth}=arcsin\left(\frac{(\psi \left(c_{j}^{2}\right)-\psi \left(c_{j}^{3}\right))\lambda }{2\pi d}\right) \end{array} \end{array}$$(12)$$\begin{array}{c} \begin{array}{c} \theta_{elevation}=arcsin\left(\frac{(\psi \left(c_{j}^{1}\right)-\psi \left(c_{j}^{3}\right))\lambda }{2\pi d}\right) \end{array} \end{array}$$
where ϕ is the phase of a complex value, $c_{j}^{i}$ is the *j*-th complex value in range-Doppler from the *i*-th receiving antenna.

In this paper, the space source signal is a narrow-band signal, the time required for the signal to pass through the array length is far less than the signal coherence time, and the signal envelope has little change in the antenna array propagation time. Assume that the signal carrier is $e^{j\omega t}$, in space, in the form of plane waves, propagating along the direction of the beam angle θ, and the signal of the receiving antenna 1 is $s_{1}\left(t\right)e^{j\omega t}$, then the signal $s_{2}\left(t\right)$ at d for receiving antenna 2 is(13)$$\begin{array}{c} \begin{array}{c} s_{2}\left(t\right)=s_{1}\left(t-t_{delay}\right)e^{j\omega t-j2\pi \frac{d}{\lambda }sin\left(\theta \right)} \end{array} \end{array}$$
where $t_{delay}$ is the delay time from antenna 1 to antenna 2, which can be ignored here *d* is the distance of the receiving antennas, and λ is the wavelength.

Therefore, the matrix signal is expressed as follows:(14)$$\begin{array}{c} \begin{array}{c} s\left(t\right)≜{\left[s_{1}\left(t\right),s_{2}\left(t\right)\right]}^{T}=s\left(t\right){\left[1,e^{-j2\pi \frac{d}{\lambda }sin\left(\theta \right)}\right]}^{T} \end{array} \end{array}$$
where *M* is the number of receiving antennas, and θ is the desired direction.

And the desired direction is expressed by the complex matrix as(15)$$\begin{array}{c} \begin{array}{c} \alpha \left(\theta \right)={\left[1,e^{j2\pi \frac{d}{\lambda }sin\left(\theta \right)}\right]}_{2\times 1}^{⊤} \end{array} \end{array}$$

The principle of digital beam forming is to stack signals in a certain direction in the same phase at the receiving end, so as to form a narrow beam in the target direction to improve the beam pointing ability in a certain direction.

Compared with classical subspace-based DOA estimation methods, such as the Music algorithm and the ESPIRT algorithm, the DBF method requires less computation and has higher robustness for fewer antenna arrays. In Figure 9, the two antennas mentioned specifically refer to Rx1 and Rx2. These two receive antennas are used to illustrate the calculation of the angle in the E-plane. For the angle calculation in the H-plane, Rx2 and Rx3 are employed based on the same principle. Due to space limitations, the details of the H-plane calculation are not repeated here.

In some recent works, the researchers have tried to represent the spatial information as the Angle-Time Map (ATM) [8,30]. As multiple channels, these spatial feature maps will be fed into various feature extractors, for instance, a CNN-based network. The spatial information was utilized in these methods to replace one dimension of RDM. However, the methods of expressing the temporal information integrated from the perspective of image recognition (adopting the idea of image recognition) require a fixed time length, which is a significant obstacle for practical applications.

This paper considers the impact of varying action length (action complexity, motion speed, duration, etc.) by decoupling spatial information from temporal information and excluding temporal information from the feature map. For each frame of the radar signal, the current gesture angle is expressed in the form of a Range-Angle Map (RAM). Spatial information is represented by a continuous time series of frames.

The procedure to extract spatially continuous multidimensional gesture features in this paper is summarized in Figure 10. First, the $N_{chan}\times RDM$ are calculated frame by frame from the raw radar data based on range-FFT and Doppler-FFT. The clutter extraction method, MTI, mentioned above, is applied to obtain background noise removal RDM. Prior to angle information estimation, the phase noise between each receiving antenna is precisely calibrated as described earlier to obtain better target angle information, which is of significant importance for sensing subtle and accurate gestures. In the zenith angle plane and in the azimuth angle plane, a 2D digital beamforming method is used to compute the AoA signal, respectively.

Generally, target detection can be implemented in the range Doppler domain or range–angle domain. In this article, the range Doppler domain is mainly used rather than the range–angle domain because the Doppler resolution is high, which helps sensing a target located in different positions and with different velocities. However, the angular resolution is very low for the used antenna array, which is not enough to resolve accurate information in the angular dimension. However, for many gestures with similar motion trajectories, such as left swipe and right swipe, their corresponding RDMs show the same process of the distance changing from large to small and then back to large, and the speed changing from negative to positive. It is difficult to distinguish these gestures solely based on the change in distance and speed. At this time, the help of angle information is indispensable and plays a critical role in assisting judgment.

As a consequence, for every frame of radar data, the data are organized into tensors with three components, as cube-𝒱, including the following: range-Doppler, range-angle (E-plane), range-angle (H-plane) (Figure 11), as depicted in Figure 12. The cube has dimensions as N×M×3, where N stands for range bins, M for Doppler samples/angle samples, and 3 for the three-dimensional attributions. Each element of the cube can be described as follows:(16)$$\begin{array}{c} \begin{array}{c} 𝒱\left(n,m,1\right)=B_{13}\left(r,\theta \right),\;𝒱\left(n,m,2\right)=B_{23}\left(r,\theta \right),\;𝒱\left(n,m,3\right)=R\left(r,v\right)\;\; \end{array} \end{array}$$
where n=1,⋯,N and m=1,⋯,M.

### 3.2. Proposed Approach for Gesture Recognition

#### 3.2.1. Overview

Recently, with the availability of large datasets, CNN-based models have proven their ability in action/gesture recognition tasks; 3D CNN architectures especially stand out for video analysis since they make use of the temporal relations between frames together with their spatial content. However, there is no clear description of how to use these models in a real-time dynamic system. With our work, we aim to fill this research gap.

In order to achieve the recognition of continuous and dynamic hand gestures, a hierarchical architecture for hand gesture consisting of a detector and a classifier is proposed in this article, illustrated in Figure 13. The continuous and dynamic hand gestures range-Doppler map sequence is detected and segmented by the lightweight detector network. In short, the detector is used as a switch signal generator that activates the classifier to recognize the detected gesture sequence. The classifier then extracts more detailed features and recognizes the gestures utilizing a relatively more extensive network.

#### 3.2.2. Trigger and Gesture Segmentation

The conventional research mostly focuses on the offline hand gesture recognition systems instead of detecting valid frames. The current gesture recognition algorithms are usually aimed at a single gesture that is pre-segmented. But in practical applications, the original data frame is usually uncut and contains multiple continuous actions. Only by detecting the beginning and end of the frame stream, hand gesture recognition is able to recognize the specific actions. The final recognition accuracy relies on both the classifier’s generalization ability and the valid input frames, which are segmented by the detector to align with the actual gesture duration.

We designed a neural network model as a detector to classify the current gesture sequence as either an action or a non-action state. To ensure the real-time running requirement, this function should repeat with a high frequency on the continuous gesture sequence in several frames (like a short window). The detector network must be lightweight, and the current sequence must have a small length so that the judgment would not be costly over time. Since the classifier recognizes the gesture sequence under the detector’s director, the detector’s output significantly affects the classifier’s final performance. This article adopts a lightweight CNN+LSTM network to achieve the gesture detection task, which can continuously work with a high frequency relying on its light weight.

To activate the classifier correctly in real-time, the detector is required to (i) have a lightweight network, (ii) have no missing detections, and (iii) have no multiple detections for one gesture attempt. For the sake of (i), a lightweight neural network was designed containing several convolutional layers and LSTM cells, as in Figure 14. Feature sizes in each layer are described in Table 3. LSTM has one hidden layer with 128 nodes. For (ii) and (iii), the detector is required to not only have a high recall rate but also give a stable output. Thus, a counter of consecutive same detection results is used to filter out rapid change.

#### 3.2.3. Gesture Classification

In Figure 13, the proposed pipeline for gesture detection is illustrated through multiple subfigures. Figure 13a depicts the range information captured by the millimeter-wave radar for each frame, aligned with the video stream shown at the top. Figure 13b illustrates the segmentation derived from the video stream, serving as a baseline for gesture detection, with the midline indicated by the pink dashed line in the legend. Figure 13c,d present the gesture detection results using the traditional threshold method and the STA/LTA method, respectively. In contrast, Figure 13e demonstrates the enhanced performance of the proposed gesture detector, achieving higher accuracy than the methods in Figure 13c,d, while effectively distinguishing between two gestures despite noise. Finally, Figure 13f extends the results of Figure 13e by incorporating 2xRAM at the input layer.

The classifier is the core part of our recognition network, which is fed the segmented sequence (like a long window) by the detector. Similarly, a CNN+LSTM network structure was adopted to serve as our classifier, though with a more significant amount of parameters to extract more complex features and more extended sequence data in time order.

The feature form refers to the reflector shape in the range-Doppler map, and the position describes the reflector’s distance and speed relative to the radar, respectively. Thus, to acquire accurate features for the current frame, the feature’s position should be considered. Unfortunately, CNN is not sensitive to the feature’s position in the image.

In study [12], the authors proposed that the CNN can implicitly learn the feature’s position in an image and figure out the positive impacts on the position learning efficiency: (i) a large number of stacked layers, (ii) a large convolution kernel size. In addition, the authors in [12] represented a novel method to promote the position learning efficiency. They added extra, hard-coded input channels that contain information about two coordinates into the original data directly. Concretely, these extra input channels could be straightforward, in which the first coordinate channel is a rank-1 matrix with its first row filled with 0, its second row with 1, its third with 2, etc. The second coordinate channel is similar, but with columns filled with constant values instead of rows. According to the previous works, we have built a CNN feature extractor with the same layer architecture in Figure 14, which has high position learning efficiency due to a larger kernel size and extra coordinate information layers. The effectiveness will be analyzed in the following experiments. Consequently, an LSTM network is attached after the CNN feature extractor to model hand gestures’ time-series information. The detailed model structure used in this article is described in Table 4.

## 4. Results

### 4.1. Data Specification and Acquisition

To train and evaluate the proposed hand gesture recognition system, we gathered a dataset: eight individuals were invited to record the hand gesture data, which contain 11 gestures. Specifically, except for 10 gestures that are the same as the setting in recent work [10], we added a new double-hand gesture (double-push). Because the FMCW mmw-radar has a low angle resolution, this new gesture can indicate the system’s spatial distinguishability. All the gestures are depicted in Figure 15.

Two of our individuals were well-trained before the data recording. Thus, they were regarded as the baseline group. Then, to make the recording procedure faster and convenient, their data were used to train the detector model in the first step. And then, this pre-trained model was used to clip and save the hand gestures automatically. Whereas the remaining six individuals in the other group only obtained an example for each gesture demonstrated by the baseline group members. They performed gestures from their habits. We found that their gestures were more varied than those of well-trained individuals.

Each individual provided 10 instances of data for each gesture. As a result, we collected 880 samples from eight participants. To ensure robust evaluation and avoid bias from individual-specific gesture patterns, we employed the Leave-One-Subject-Out (LOSO) cross-validation for model assessment. In this evaluation protocol, the data from seven participants were used for the training, while the remaining participants’ data served as the test set. This process was repeated eight times, with each participant serving as the test subject once, providing a comprehensive evaluation of the system’s generalization capability across different users.

### 4.2. Detection Performance

Since the detection features are much more straightforward than the classification features, the detector model was trained on the baseline group data. Specifically, the CNN-based detector was trained using the Adam optimizer with a fixed learning rate of $1\times 10^{-4}$. Under the LOSO cross-validation protocol, the detector achieves an average accuracy of 98.0% across all eight test participants. The confusion matrices are represented in Figure 16.

### 4.3. Classification Performance

The classifier model was trained with the same optimizer and the same learning rate as the detector. No batch normalization technique was adopted due to the various data lengths. In Figure 17a,b, two confusion matrices for classifier performance are plotted. The matrices represent the average performance across all LOSO cross-validation folds, showing both training performance and cross-validation results. According to the work [10], a feature cube with the CNN method was evaluated. The performance results are considered in this article since the same gestures are adopted, so that it is allowable to compare these two works. The detailed result is listed in Table 5. Our proposed approach has a higher average accuracy 93.87% across all eight test participants.

Figure 17b presented the recognition accuracies of the “Clockwise” and “Push” gestures are 0.80 and 0.87, respectively. These values are notably lower than the average accuracy, indicating that these two gestures are corner cases in our gesture recognition system. Detailed data analysis indicates that misclassifications primarily stem from variations in gesture execution by participants. When participants perform the “Clockwise” and “Push” gestures, their actual movements deviate from the standard ones. This deviation leads to confusion between these two gestures, thereby reducing the recognition accuracy of the model. To tackle this issue, in the future, we plan to collect additional data specifically for the “Clockwise” and “Push” gestures. The new data will be expected to be more diverse, encompassing a broader spectrum of possible gesture deformations. With this augmented dataset gathered, we intend to retrain our model to enhance its capability to differentiate between these similar gestures and optimize its overall performance.

To further contextualize our system, Table 6 compares the design characteristics, such as radar platform, IoT integration, and gesture types, with other similar works.

### 4.4. Ablation Study on Multi-Feature Fusion

Table 2 presents a comprehensive ablation study evaluating the effectiveness of our proposed multi-feature fusion approach. The results demonstrate the progressive improvement achieved by incorporating additional spatial features into the gesture recognition framework.

The baseline configuration using only Range-Doppler Map (RDM) with a single-channel CNN+LSTM architecture achieves 94.9% training accuracy and 90.3% test accuracy. However, the relatively high loss values (0.182 for training and 0.249 for testing) indicate the potential for overfitting and a limited generalization capability. The gap between training and test performance (4.6%) suggests that relying solely on RDM features may not capture sufficient spatial information for robust gesture recognition across different users.

When incorporating one Range-Angle Map (RDM+1xRAM) with a 2-channel CNN+LSTM, we observe modest improvements: the training accuracy increases to 95.1% and test accuracy to 91.4%. More importantly, the loss values decrease substantially (0.114 for training, 0.193 for test), indicating better model convergence and reduced overfitting. The training-test gap narrows to 3.7%, demonstrating enhanced generalization.

The most significant performance gains are achieved with the complete multi-feature fusion approach (RDM+2xRAM) using a 3-channel CNN+LSTM architecture. This configuration attains 98.5% training accuracy and 93.1% test accuracy, representing a 2.8% improvement in test performance compared with the baseline. The reduction in loss values (0.028 for training, 0.059 for test) indicates optimal model convergence, while the reduced training-test gap (5.4%) demonstrates robust generalization capability despite the increased model complexity.

The ablation results validate our hypothesis that spatial information from multiple Range-Angle Maps significantly enhances gesture recognition performance. The progressive improvement from RDM-only to RDM+2xRAM confirms that angular information from both E-plane and H-plane provides complementary spatial features that are crucial for distinguishing between gestures with similar range-velocity profiles but different spatial trajectories. This is particularly important for our 11-gesture vocabulary, which includes spatially similar gestures like “left” “right” and “clockwise” movements.

### 4.5. Real-Time Evaluation

Since the signal processing, gesture detection, and classification tasks can be executed in parallel, the multiprocessing technique is used to speed up the whole system. During the training phase, a PC with Intel low-voltage chip i7-8565U @ 1.80 GHz CPU and NVIDIA GTX 1080 GPU was utilized for model optimization.

The quantized CNN classifier demonstrates exceptional efficiency for edge deployment, achieving a compact model size of 2.1 MB through TensorFlow Lite’s full integer quantization (FP32 to INT8) and depthwise separable convolution optimization. Notably, this quantization process did not cause any observable degradation in detection accuracy compared with the full-precision model. As shown in Figure 18, when deployed on Raspberry Pi 4 (Figure 4), the model maintains real-time performance with a mean inference latency of 15.4 ms (±0.6 ms) when processing individual samples, making it highly suitable for resource-constrained clinical environments. Table 7 summarizes the complexity analysis of the gesture recognition system, including GFLOPs, model size and average runtime by considering the state of the art. The GFLOPs of our model (0.24) were calculated by the built-in function in TensorFlow.

## 5. Discussion

### 5.1. Deployment and Implementation in Ubiquitous Clinical Application Environments

To validate the practical viability of our system for widespread adoption in real-world healthcare settings, we deployed the proposed FMCW radar-based gesture recognition framework within a smart hospital ward environment. This deployment served as a representative testbed for demonstrating the system’s capability to provide ubiquitous [33], contactless patient interaction and convenient environmental control through intuitive hand gestures. The goal was to realize a smart healthcare scenario that is widely applicable across various clinical settings [34].

Our field tests within the clinical environment further demonstrated the system’s versatility and applicability to other essential and ubiquitous clinical controls. Specifically, patients could control the hospital room light switch using simple up-and-down wave gestures, in addition to summoning nursing assistance with another predefined gesture. These two functions—bed elevation (via push/pull swipe) and light control (via left/right wave)—represent fundamental, widely applicable interactions required in virtually all hospital rooms. Implementing these via gesture provides patients with a contactless, hygienic, and convenient method of control, directly contributing to a smart healthcare environment by reducing reliance on physical contact surfaces and minimizing the need for direct staff assistance for routine tasks.

### 5.2. Deployment in Specific Clinical Scenarios for Patients with Language Communication Disorders

Beyond facilitating the ubiquitous control of the clinical environment, our radar-based gesture recognition system holds profound potential as a critical communication tool for patients experiencing challenges with verbal expression [35]. This encompasses a wide spectrum of individuals affected by diverse conditions, including neurodevelopmental disorders (such as autism spectrum disorder, cerebral palsy, or developmental delays), acquired neurological conditions (like stroke, traumatic brain injury, or neurodegenerative diseases such as Parkinson’s or Alzheimer’s), and temporary or situational limitations resulting from post-surgical recovery (e.g., oral/facial surgeries), respiratory illnesses (e.g., severe asthma or chronic obstructive pulmonary disease exacerbations requiring intubation), acute injuries, or certain mental health conditions in Figure 19 [36,37,38]. These varied circumstances can significantly impede a patient’s ability to communicate basic needs, discomfort, or responses, creating significant barriers to effective care.

In such clinical scenarios where traditional verbal communication is difficult or impossible, our system offers a crucial alternative modality. By associating simple, intuitive hand gestures with specific, essential expressions or commands, the system enables patients to convey vital information in real-time. For instance, a patient can easily signal the presence or severity of pain with a designated gesture, provide affirmative (“Yes”) or negative (“No”) responses to clinical questions, or communicate fundamental needs such as “I am hungry.” “I am thirsty.” or “I need to use the restroom.”.

This real-time, contactless interface provides a non-verbal, direct communication channel. It facilitates a clear, one-to-one mapping between the patient’s gesture and the intended message, allowing for efficient and immediate translation of their needs or state into actionable information for healthcare staff. This capability is particularly valuable in settings where speech is limited or compromised, bridging the communication gap and enhancing responsiveness. Furthermore, the inherent simplicity and low cognitive load associated with performing basic hand gestures make the system accessible even to patients who may have concurrent cognitive or motor impairments.

Implementing this gesture-based communication system within clinical workflows significantly enhances patient-centered care. It empowers patients to actively participate in managing their comfort and needs, alleviates the frustration and isolation often associated with communication barriers, and ensures more timely and accurate clinical responses to urgent requirements. Its versatility makes it applicable across various hospital settings, from intensive care units and rehabilitation wards to long-term care facilities, profoundly benefiting individuals with speech and language challenges.

Future efforts will focus on tailoring gesture sets to the specific requirements of different patient populations and clinical contexts, ensuring maximum usability, clarity, and recognition accuracy. Continuous, collaborative development with healthcare professionals, including speech-language pathologists, nursing staff, and patients themselves, will be essential for the iterative refinement and successful adoption. Ultimately, deploying radar-based gesture recognition as a dedicated communication aid holds the potential to fundamentally transform patient-provider interaction for those with communication impairments, promoting dignity, independence, and fostering a more inclusive and accessible healthcare environment.

## 6. Conclusions

This study demonstrates a robust, contactless human–machine interaction system based on FMCW radar integrated into medical sensor platforms, tailored for clinical environments. Through a dual-stage processing approach utilizing adaptive energy thresholding and clutter suppression, our system achieves a high detection recall of 98%, ensuring reliable gesture detection even in noisy settings. Coupled with an attention-enhanced neural network and an efficient multi-feature fusion scheme—including velocity, AoA, and spatial-temporal features—the system attains a generalization classification accuracy evaluated via Leave-One-Subject-Out (LOSO) protocol of 93.87% across 11 gesture categories. These advancements outperform traditional methods by 7–12%, offering richer data representation with minimal computational overhead.

Implemented on embedded hardware such as the Raspberry Pi 4, our system operates at 28 FPS and benefits from a reduction in computational load, enabling real-time, low-power deployment suitable for portable medical devices or wearables. Its privacy-preserving, hygienic, and through-material operation makes it ideal for sterile environments, remote patient monitoring, device control, and assistive applications.

Building on these results, future work will focus on expanding datasets to encompass diverse users and environments, enhancing clutter suppression for low Signal-to-Noise Ratio and extended-range scenarios, and exploring sensor fusion modalities. Long-term clinical validation will be essential to ensure safety, robustness, and reliability in real-world healthcare settings. Additionally, we plan to transition from Raspberry Pi to ESP32-based platforms, facilitating the tighter integration of sensor chips while maintaining real-time performance and energy efficiency.

Furthermore, we aim to deploy the system in two strategic pathways: (i) ubiquitous clinical environments with gesture-controlled room functions such as lighting and bed adjustments, and (ii) specialized scenarios for patients with communication disorders, employing predefined gesture-to-semantic mappings to facilitate essential care communication. These developments will advance the next generation of contactless HCI solutions, spanning routine clinical operations and critical patient-care interactions, ultimately promoting safer, more accessible, and more efficient healthcare delivery.

## Figures and Tables

**Figure 1 sensors-25-04169-f001:**
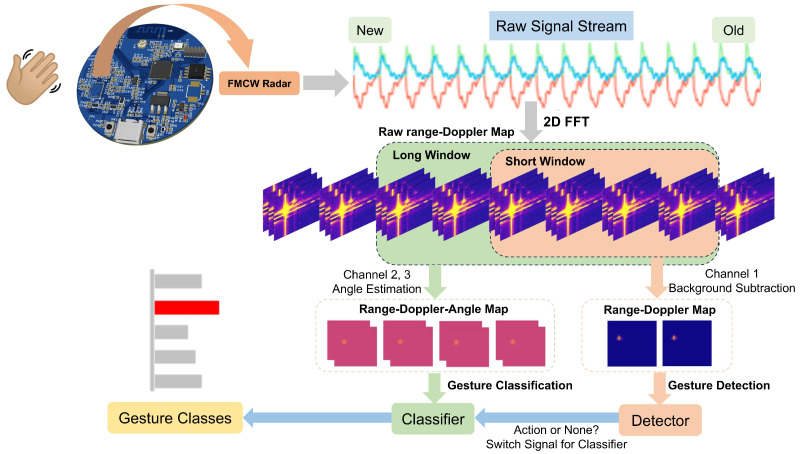
System architecture overview.

**Figure 2 sensors-25-04169-f002:**
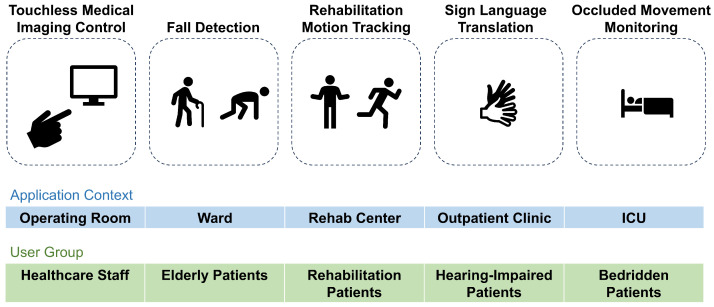
Radar-based gesture recognition in healthcare applications.

**Figure 3 sensors-25-04169-f003:**
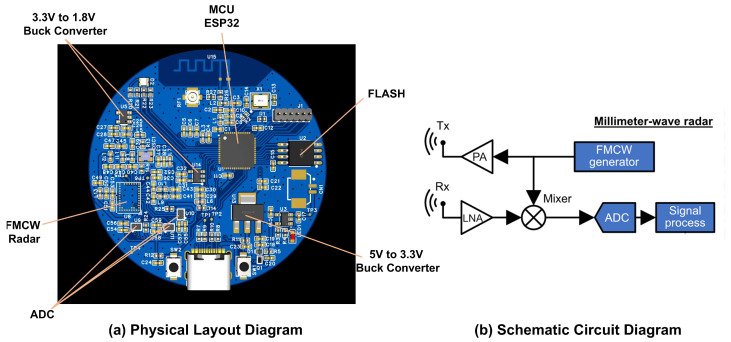
System Layout: (**a**) physical layout diagram, (**b**) schematic circuit diagram.

**Figure 4 sensors-25-04169-f004:**
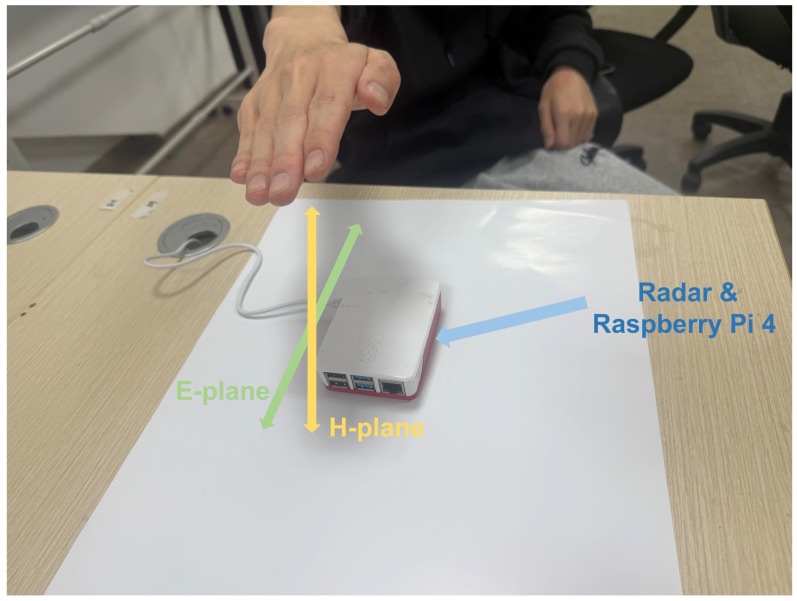
Experiment scenario for real-time gesture recognition.

**Figure 5 sensors-25-04169-f005:**
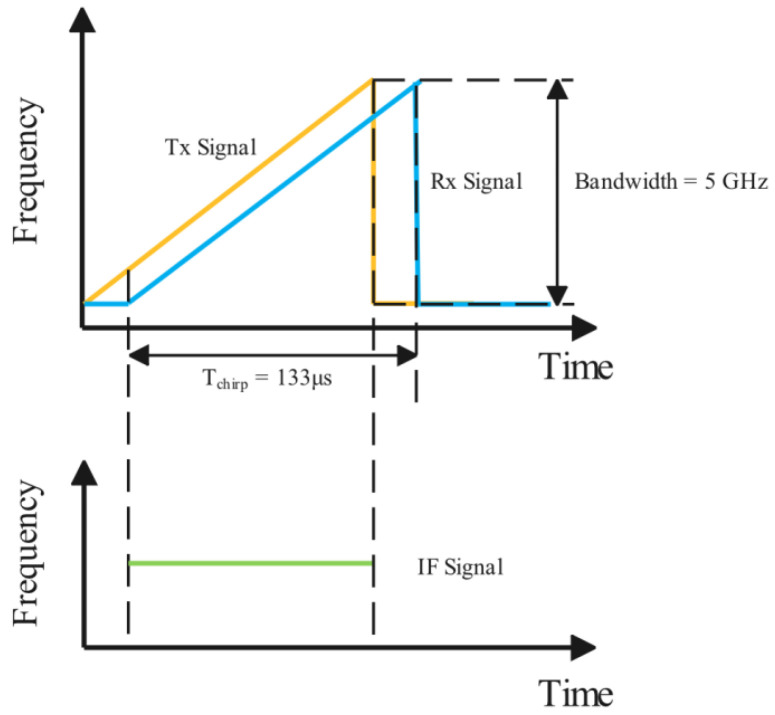
FMCW waveform.

**Figure 6 sensors-25-04169-f006:**
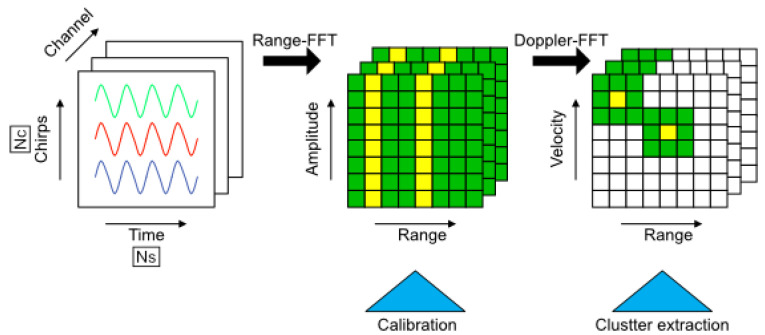
The preprocessing chain for extracting the Doppler information of possible targets for a standard multichannel FMCW radar. Different color lines in the “Channel” stage represent signals from different channels. Green and yellow cells in the “Range-FFT” and “Doppler-FFT” stages indicate regions with higher signal strength, with yellow cells highlighting potential targets.

**Figure 7 sensors-25-04169-f007:**
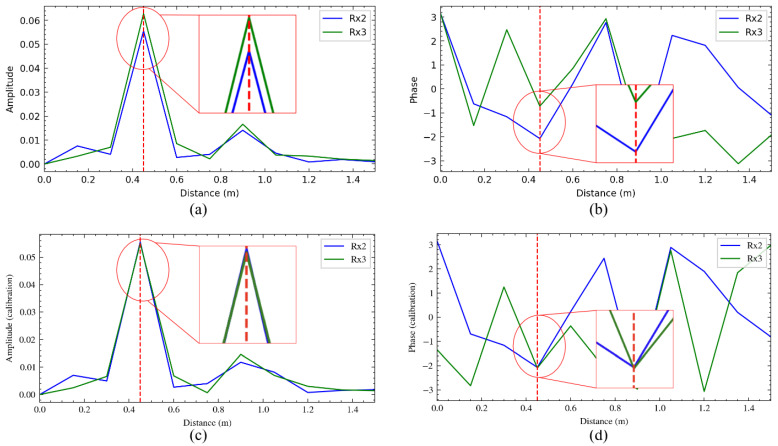
Range profile: (**a**) uncalibrated amplitude, (**b**) uncalibrated phase, (**c**) calibrated amplitude and (**d**) calibrated phase. The red dashed line indicates the approximate range of the target. The insets show a zoomed-in view of the peak response for each plot.

**Figure 8 sensors-25-04169-f008:**
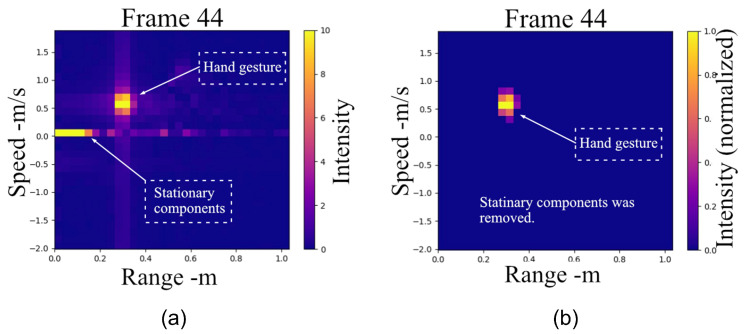
Background noise subtraction: (**a**) Before background subtraction, (**b**) After background subtraction.

**Figure 9 sensors-25-04169-f009:**
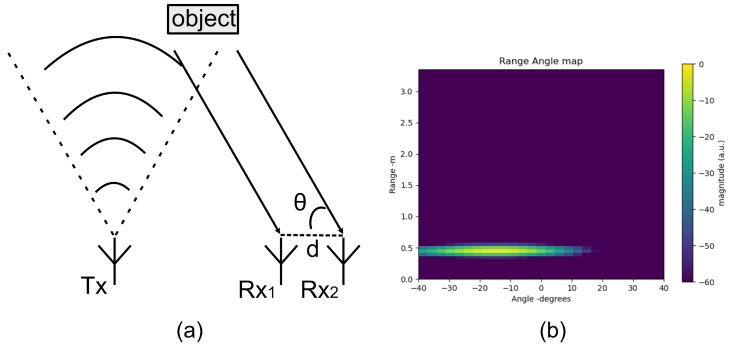
DBF method: (**a**) principle of DBF, (**b**) range angle map.

**Figure 10 sensors-25-04169-f010:**
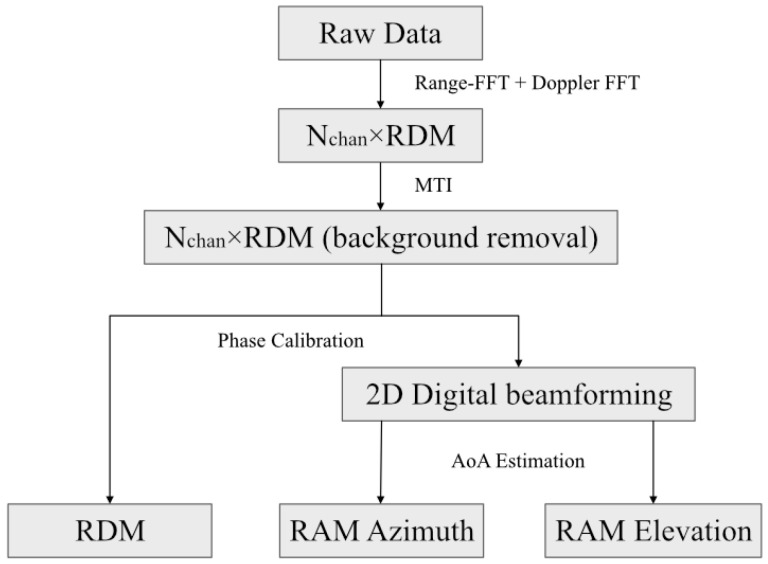
Procedure to extract multidimensional gesture features.

**Figure 11 sensors-25-04169-f011:**
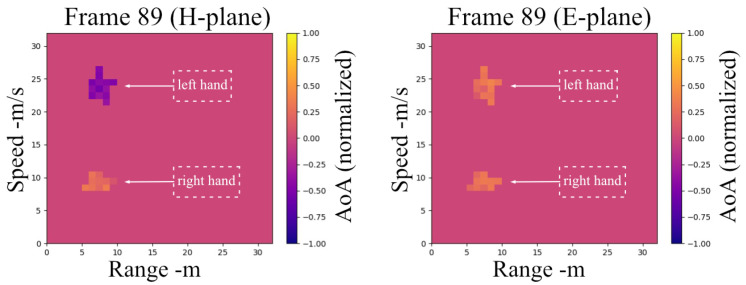
RDAMs for H-plane and E-plane while two hands are moving above the radar.

**Figure 12 sensors-25-04169-f012:**
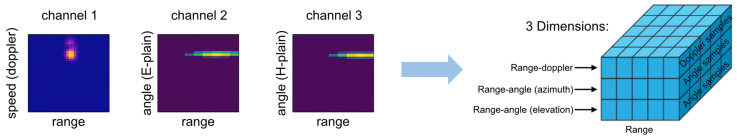
Structure of Feature Cube.

**Figure 13 sensors-25-04169-f013:**
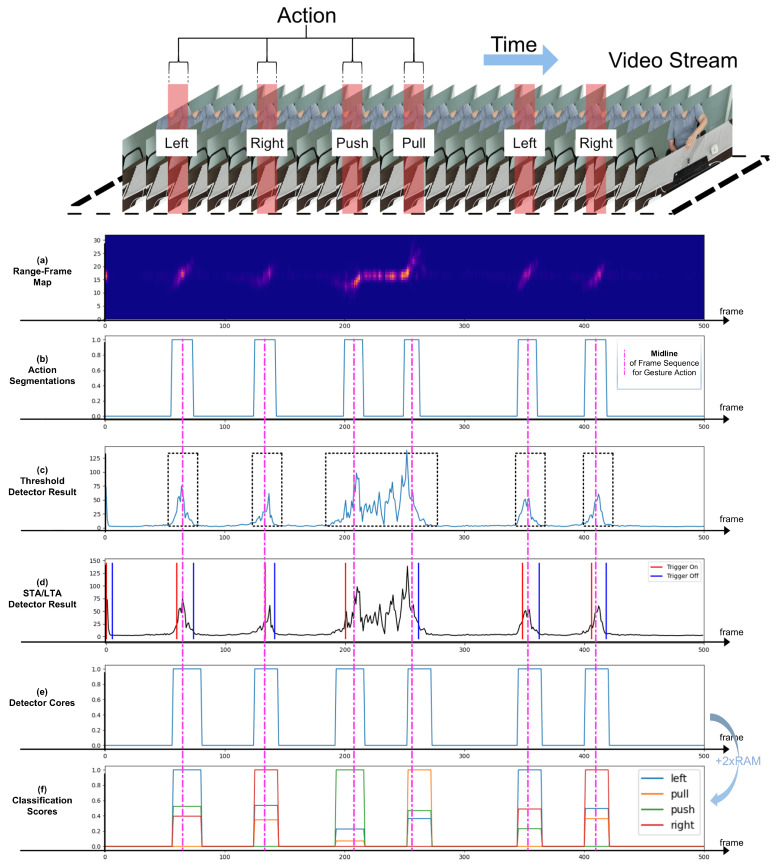
Illustration of the proposed pipeline for real-time gesture recognition.

**Figure 14 sensors-25-04169-f014:**
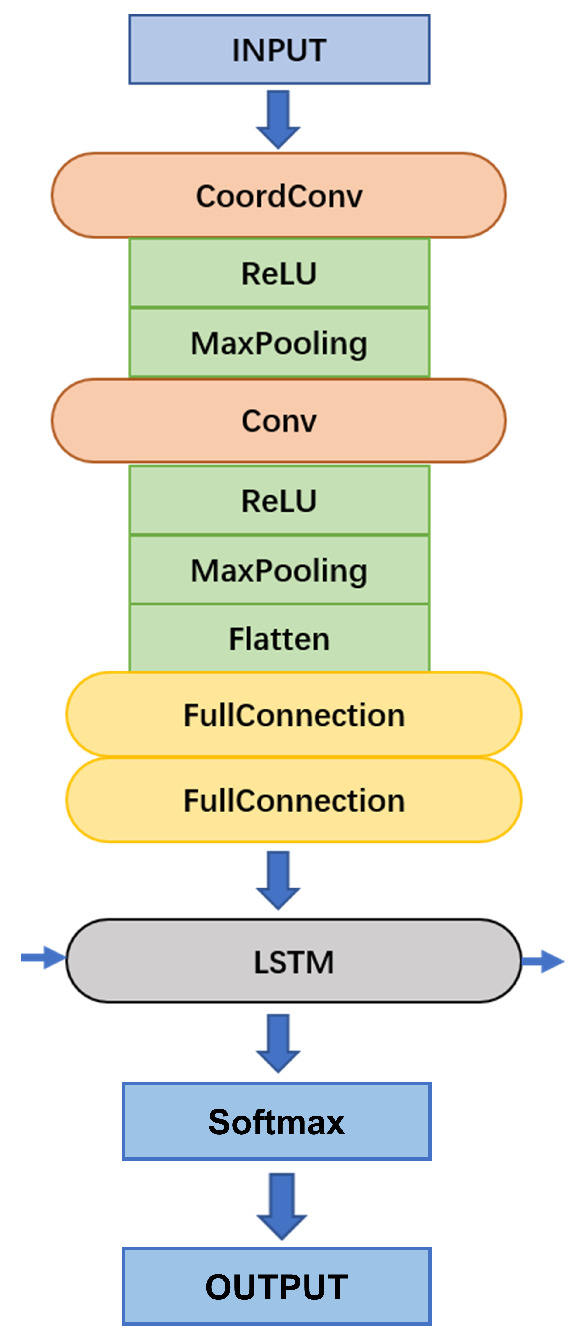
Neural network layer structure.

**Figure 15 sensors-25-04169-f015:**
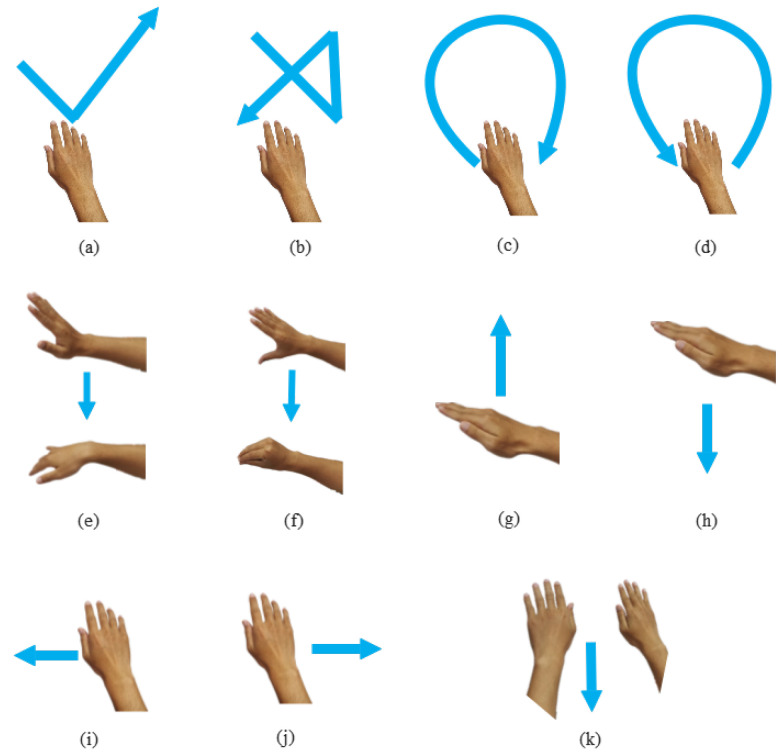
Gesture type (**a**) Check. (**b**) Cross. (**c**) Clockwise. (**b**) Anti-clockwise. (**e**) Move fingers. (**f**) Pinch. (**g**) Pull. (**h**) Push. (**i**) Left. (**j**) Right. (**k**) Double push.

**Figure 16 sensors-25-04169-f016:**
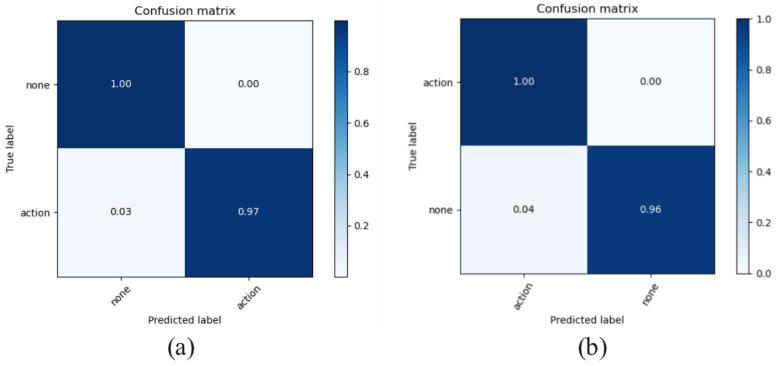
Confusion matrix of detector: (**a**) Train set, (**b**) cross-validation.

**Figure 17 sensors-25-04169-f017:**
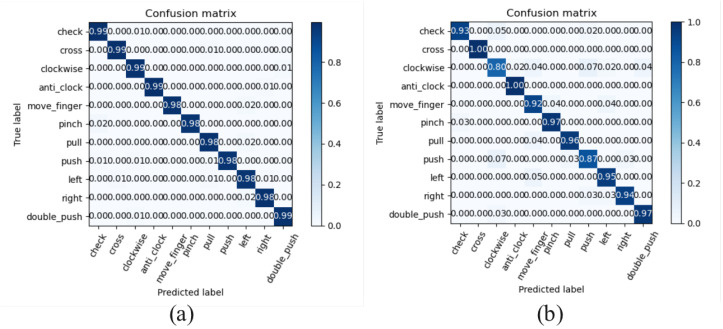
Confusion matrix of classifier: (**a**) Training set, (**b**) Cross-validation.

**Figure 18 sensors-25-04169-f018:**
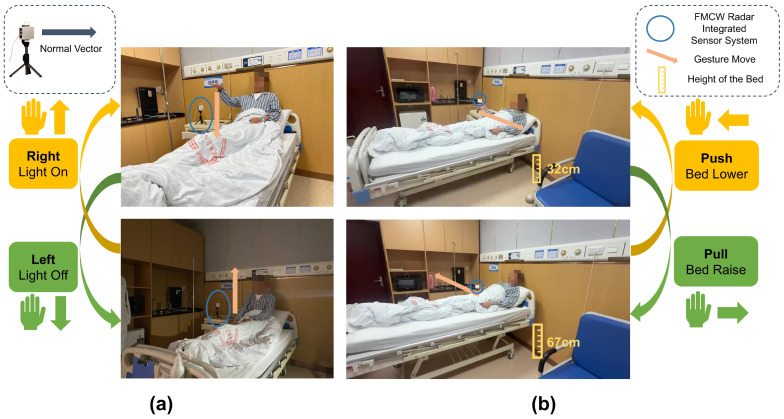
Clinical implementation and application: (**a**) Gesture-controlled Light On/Off, (**b**) gesture-controlled bed adjustment.

**Figure 19 sensors-25-04169-f019:**
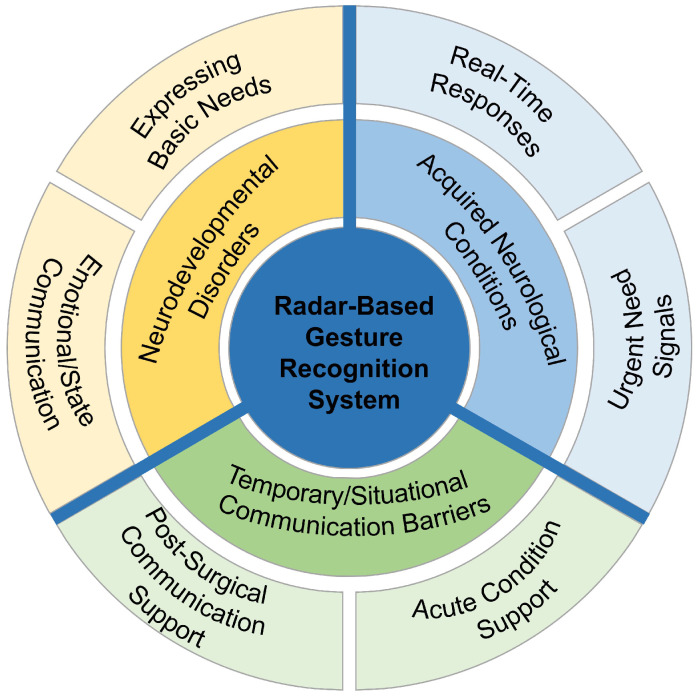
Radar gesture recognition TOC for patient communication.

**Table 2 sensors-25-04169-t002:** Comparison of classification results of three multi-feature fusions.

Feature Type	Classification Method	Average Accuracy		Average Loss	
		**Training Set**	**Test Set**	**Training Set**	**Test Set**
RDM	1-channel CNN + LSTM	94.9%	90.3%	0.182	0.249
RDM + 1 × RAM	2-channel CNN + LSTM	95.1%	91.4%	0.114	0.193
RDM + 2 × RAM	3-channel CNN + LSTM	98.5%	93.1%	0.028	0.059

**Table 3 sensors-25-04169-t003:** Layer architecture of the detector.

Layer	Input	Output	Kernel
CoordConv	1 × 32 × 32	4 × 30 × 30	3 × 3
MaxPool	4 × 30 × 30	4 × 15 × 15	2 × 2
Conv	4 × 15 × 15	8 × 13 × 13	3 × 3
MaxPool	8 × 13 × 13	8 × 6 × 6	2 × 2
Flatten	8 × 6 × 6	288	–
FC	288	128	–
FC	128	64	–
LSTM	64	2	–

**Table 4 sensors-25-04169-t004:** Layer architecture of the classifier.

Layer	Input	Output	Kernel
CoordConv	1 × 32 × 32	32 × 30 × 30	3 × 3
MaxPool	32 × 30 × 30	32 × 15 × 15	2 × 2
Conv	32 × 15 × 15	64 × 13 × 13	3 × 3
MaxPool	64 × 13 × 13	64 × 6 × 6	2 × 2
Flatten	64 × 6 × 6	2304	–
FC	2304	768	–
FC	768	256	–
LSTM	64	11	–

**Table 5 sensors-25-04169-t005:** Classification accuracy in % of each gesture by different gesture recognition systems.

Method	Avg.Acc.	(a)	(b)	(c)	(d)	(e)	(f)	(g)	(h)	(i)	(j)	(k)	F1 Score
[10]	91.45	88.24	88.33	95.83	85.07	98.18	87.23	73.53	98.08	100.0	100.0	–	88.58
Proposed method	93.87	93.07	100.0	80.33	100.0	92.18	97.34	96.18	87.24	95.03	94.23	97.01	93.81

**Table 6 sensors-25-04169-t006:** Comparison with similar works.

Reference	[10]	[31]	[32]	[16]	[15]	This Work
**Radar Platform**	BGT60TR13C	IWR1443	AWR1642	BGT60TR13C	AWR1243	BGT60TR13C
**IoT Design**	Yes	No	No	No	No	Yes
**Working Frequency**	60 GHz	77 GHz	77 GHz	60 GHz	77 GHz	60 GHz
**Gesture Length**	Variable (threshold)	Fixed	Fixed	Variable	Fixed	Variable
**Gesture Types**	12 gestures	10 gestures	8 gestures	8 gestures	9 gestures	11 gestures

**Table 7 sensors-25-04169-t007:** Complexity analysis of the gesture recognition system.

Method	GFLOPs	Size	Average Runtime
[10]	0.26	4.18 MB	33.15 ms
Proposed method	0.24	2.1 MB	15.4 ms (±0.6 ms)

## Data Availability

The original contributions presented in this study are included in the article. Further inquiries can be directed to the corresponding author(s).

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
