# Peer review of "Real-Time Hand Gesture Recognition in Clinical Settings: A Low-Power FMCW Radar Integrated Sensor System with Multiple Feature Fusion"

_sensors, 2025, doi:10.3390/s25134169_

Round 1
Reviewer 1 Report
Comments and Suggestions for Authors
1. There is no explanation for Figure 5, such as how to suppress clutter from the range Doppler plot in Figure 5?
2. Please confirm that the legends in Figure 6 are correct. Are the colors of the lines displayed by RX2 and RX3 in (c) and (d) exactly opposite to those shown in (a) and (b)?
3. Some formula variables are not explained, such as Equations (3), (4), and (12).
4. In some parts of the paper, it is described as having three receiving antennas, while in others it is described as having two. Please keep the descriptions consistent.
5. Please provide more explanations for Equation (15). Why can clutter be estimated and removed?
Reviewer 2 Report
Comments and Suggestions for Authors
Real-Time Hand Gesture Recognition in Clinical Settings: A Low-Power FMCW Radar Integrated Sensor System with Multiple Feature Fusion
The article proposes an end-to-end framework for a Radar sensor-based gesture recognition algorithm. The authors have implemented an embedded system that captures the gestures and then uses a classification model to identify gestures alongside their fine-grained classes. In addition, the authors discuss the applicability of the system in real-world settings, which makes their contributions significant. I have comments about the deep learning model development from my expert end.
From Section 2.3. Proposed Approach for Gesture Recognition. The authors propose two architectures: gesture recognition and fine-grained classification. Table 2 presents the layer architecture of the first model, and it has an output shape of 64, which is incomplete for a binary classification model (it should be one or two). Similarly, in Table 3, the authors have not given full details. The model has 256 as the output dimension, but can there be such a large number of gestures to be detected?
“As a result, we collect 880 samples, which are then divided randomly to the train set and test set in a proportion of 325 0.75:0.25. The network models in proposed approach will be trained and validated by these 326 two sets, respectively.” I am concerned about the evaluation protocol and the number of subjects involved—presumably eight.
- Firstly, if model complexity was a concern, it’s unclear why classical machine learning models were not considered.
- The proposed LSTM-CNN model is not lightweight. The mention of "128 nodes per layer" is vague—does this refer to the LSTM’s hidden dimension? If so, how many LSTM layers were used?
- Additionally, a fully connected layer with dimensions 288×128 introduces a large number of trainable parameters, which seems excessive given the dataset size of only 880 samples.
- The evaluation protocol lacks robustness. I strongly suggest using Leave-One-Subject-Out cross-validation to better assess generalization in realistic, subject-independent settings.
Given the limited data, I doubt deep learning is adequately supported. Without a more rigorous evaluation, the contribution in terms of DL remains weak. Classical machine learning models may be more suitable here, or features directly extracted from pre-trained CNN models.
Reviewer 3 Report
Comments and Suggestions for Authors
All tables and figures must be called and explained in the text of paper. For instance, there is not reference for Table I.
A paragraph about the rest of the paper must be inserted at the end of the introduction.
Section "2.1. Related works" does not belongs to Methods and Materials.
Indentation (space) should be removed when explaining variables in equations.
Training, validation, and testing results and discussion must be presented.
Comparison in particular performances must be inserted, such as F1-score or Geometric-mean.
Complexity analysis must be added by considering the state of the art.
Abbreviations in your section should be listed alphabetically.
Reviewer 4 Report
Comments and Suggestions for Authors
This manuscript presents a technically sound and timely contribution to the field of radar-based human-machine interaction (HMI), specifically tailored for clinical settings. The authors propose a real-time hand gesture recognition system based on low-power FMCW radar, emphasizing edge-device compatibility, privacy preservation, and contactless interaction. The system architecture—featuring a dual-stage detector-classifier pipeline and a multi-feature fusion framework—is compelling, and the manuscript is well-structured and written clearly. The experimental validation, both in terms of classification metrics and deployment on a Raspberry Pi platform, supports the practicality and applicability of the proposed approach. The paper also effectively connects the technical solution to real-world medical challenges, including use cases for patients with communication impairments. However, a few issues need to be addressed to enhance the manuscript’s clarity, scientific rigor, and completeness. I recommend acceptance with minor revisions as outlined below. These improvements will enhance the clarity, robustness, and applicability of an already well-executed study.
1- While the paper references several similar systems, the distinctive contributions compared to recent studies (e.g., [15], [36]) should be made more explicit.
I suggest adding a summarized comparison table highlighting the key differences in radar configuration, feature fusion, system latency, and gesture diversity.
2- The benefit of the full 3-channel fusion (RDM + 2xRAM) is evident, but the individual contributions of each component are not isolated.
authors should provide an ablation study showing model performance for each feature set (RDM only, RDM+RAM(H), RDM+RAM(E)).
3- No analysis is provided on which gestures were most frequently misclassified.
authors should Add a brief paragraph discussing common misclassifications and gesture similarities (e.g., push vs. double push), ideally supported by confusion matrix details.
4- It’s unclear how the model generalizes to unseen users, especially considering user variability in gesture performance. If available, include a leave-one-subject-out validation test or acknowledge this as a limitation and future work.
5- Improve clarity of figures (e.g., Figs. 6, 10, 13). Add annotations or brief captions explaining the significance of each subfigure. Ensure consistent formatting and labeling across all visual elements.
6- Some references are repeated and formatting is inconsistent (e.g., [15] appears in multiple sections). Ensure all references conform to MDPI citation style.
7- The transition plan from Raspberry Pi to ESP32 is briefly noted. authors should Add details on how model quantization or compression will be adapted for ESP32 deployment.
8- A few instances of awkward phrasing exist (e.g., “the current gesture sequence between action or no action status”). authors are recommended to consider a light language polish for clarity and fluency.
Round 2
Reviewer 3 Report
Comments and Suggestions for Authors
Paper can be accepted in current form.